# Short window attention enables long-term memorization

**Loïc Cabannes[1,3], Maximilian Beck[2,*], Gergely Szilvasy[1], Matthijs Douze[1],
Maria Lomeli[1], Jade Copet[1], Pierre-Emmanuel Mazaré[1], Gabriel Synnaeve[1], Hervé Jégou[1]**

[1]Meta FAIR     [2]Johannes Kepler University Linz
[3]Ecole Normale Supérieure Paris Saclay, Paris Cité University, Inria Paris

[*]Work done during an internship at Meta FAIR

## ABSTRACT

Recent works show that hybrid architectures combining local sliding window attention layers and global attention layers outperform either of these architectures taken separately. However, the impact of the window length and the interplay between local layers and global layers remain under-studied. In this work, we first analyze the interaction between short and long term memory by considering SWAX: a hybrid architecture consisting of sliding-window attention and xLSTM linear RNN layers.

A counter-intuitive finding is that larger sliding windows hurts the long-context performance. In fact, short window attention encourages the model to better train the long-term memory of the xLSTM as it cannot rely on the local softmax attention mechanism for long context-retrieval. We also validate our findings on local-global architectures alternating short window and full attention layers: the short layers should be small in order not to hinder the usefulness of the long layers.

However, employing too small sliding windows is detrimental even for short-context tasks, which could be solved with information from moderately larger sliding windows otherwise. Therefore, we train hybrid architectures by stochastically changing the sliding window size, forcing the model to leverage both the short term window and the long-term memory. Training with stochastic window sizes significantly outperforms regular window attention both on short and long-context problems.

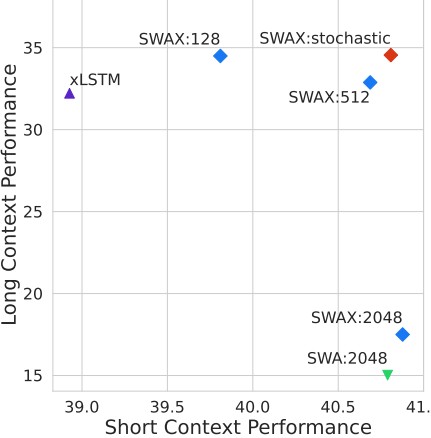

Figure 1: Short (average score across benchmarks) vs long context performance for 1.4B xLSTM, SWA (sliding window attention) and SWAX with different sliding window sizes.

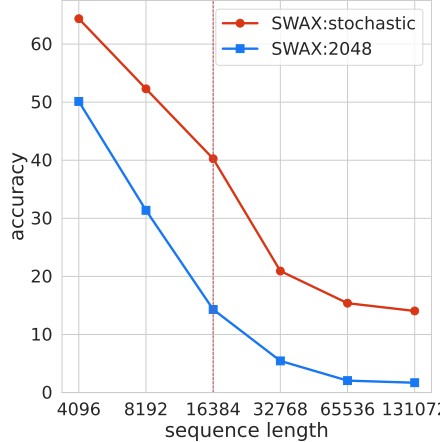

Figure 2: RULER Needle-In-A-Haystack accuracy of a 1.4B SWAX model with a fixed sliding window size of 2048 vs our method using a stochastic window size of 128/2048.

# 1 INTRODUCTION

Memory is a core concept in Neural Network Architectures (Zhong et al., 2025). Modern LLMs based on softmax attention have a working memory in the form of the key-value (KV) Cache and yield state-of-the-art long-context performance. This working memory expands indefinitely as the sequence length grows, incurring a linear growth in both compute and memory to generate each new token. With such an unbounded compute cost, current models become prohibitively expensive for in-context learning on long sequences such as codebases and long reasoning traces.

On the other hand, recurrent neural networks (RNNs) like State Space Models (SSMs) (Gu & Dao, 2024) or variants of linear attention (LA) (Katharopoulos et al., 2020) maintain and iteratively update a hidden state. Through input-dependent update rules, RNNs manage to decide whether to keep previous or add new information (Hochreiter & Schmidhuber, 1997; Chung et al., 2014). In this way, the compute and memory cost is constant and independent of the sequence length. This allows models to learn at test time from large sequence lengths and reason without a specific token limit. Recently, these linear RNNs have been generalized in the context of online learning (Liu et al., 2024; Sun et al., 2025). However, the recall ability of linear RNNs remains inferior to that of Transformers (Fu et al., 2023). This shortfall has hindered their adoption in favor of global attention-based architectures, which remain the current state-of-the-art architecture for language and code models (Jain et al., 2024).

Recent works like those of De et al. (2024), Ren et al. (2025), Dong et al. (2024) and Arora et al. (2025) have aimed at combining the advantages of softmax attention and Linear Attention into hybrid architectures (Wang et al., 2025). Following this line of research, in this paper, we study hybrid architectures, which combine linear RNNs and *sliding window* attention – both components with fixed maximum state size and thus fixed compute cost per token.

In this context, we make the following contributions:

1. We study the impact of the sliding window length on a wide range of tasks, encompassing validation perplexity, short-context reasoning, common sense benchmarks, and long-context modeling tasks;

2. We show that, contrary to previous belief, for hybrid architectures that interleave sliding window attention and linear RNNs, *longer* sliding windows actually *hurt* performance in long-context retrieval tasks compared to using *shorter* windows;

3. We present a training strategy based on a stochastic window size that achieve the best of both worlds: we attain the long-context performance enabled by short windows. At the same time the performance is on par for short-context and reasoning tasks associated with longer windows.

# 2 BACKGROUND

The attention mechanism handles sequences of key vector $\mathbf{k}_t \in \mathbb{R}^{d_{qk}}$ and value vectors $\mathbf{v}_t \in \mathbb{R}^{d_v}$. A fundamental perspective proposed by Katharopoulos et al. (2020) is that all forms of attention update a matrix memory by adding to it the outer product of key vector $\mathbf{k}_t \in \mathbb{R}^{d_{qk}}$ and value vectors $\mathbf{v}_t \in \mathbb{R}^{d_v}$. This holds, provided that we can apply a vector mapping to each of these vectors. Then, to read from the memory, a query $\mathbf{q}_t \in \mathbb{R}^{d_{qk}}$ is compared to the previous keys using a similarity metric, usually the inner product $\langle \mathbf{q}, \mathbf{k} \rangle$. In order to improve the accuracy of subsequent retrieval operations, a pre-processing feature mapping $\phi$ is applied to the keys and queries. Defining the memory tensor as

$$\mathbf{H}_t \;=\; \sum_{t=1}^{S} \phi(\mathbf{k}_t)\,\mathbf{v}_t^{\top} \;\in\; \mathbb{R}^{d_{qk} \times d_v}, \qquad \mathbf{z}_t \;=\; \sum_{t=1}^{S} \phi(\mathbf{k}_t) \;\in\; \mathbb{R}^{d_{qk}}, \tag{1}$$

a normalized read is performed as

$$\mathbf{y}_t \;=\; \frac{\phi(\mathbf{q}_t)^{\top}\mathbf{H}_t}{\phi(\mathbf{q}_t)^{\top}\mathbf{z}_t} \;=\; \frac{\sum_{i \leq t}\langle \phi(\mathbf{q}_t), \phi(\mathbf{k}_i) \rangle\,\mathbf{v}_i}{\sum_{i \leq t}\langle \phi(\mathbf{q}_t), \phi(\mathbf{k}_i) \rangle}. \tag{2}$$

**Linear attention.** Equation (1) shows that if the kernel $\phi$ is a finite-dimensional mapping, then the feature-mapped keys as well as the memory tensor are also finite-dimensional and can be materialized and cached for future retrievals $(\mathbf{H}_t, \mathbf{z}_t)$ in *constant* memory:

$$\mathbf{H}_t \leftarrow \mathbf{H}_{t-1} + \phi(\mathbf{k}_t)\,\mathbf{v}_t^\top, \qquad \mathbf{z}_t \leftarrow \mathbf{z}_{t-1} + \phi(\mathbf{k}_t). \tag{3}$$

All the keys and values are thus stored in constant memory. The per-token read cost is $O(d_{qk} \times d_v)$. Importantly, it does not depend on the sequence length $S$.

**Softmax attention (SA).** Katharopoulos et al. (2020) show that softmax attention can be seen as performing the attention operation defined in Equations 1 and 2, i.e., as writing outer products between keys and values in a memory. In such a case, the keys and queries undergo an infinite-dimensional feature mapping induced by the exponential kernel in softmax attention. Compared to linear attention, an infinite-dimensional exponential feature map reduces the interference between the stored keys and yields an improved retrieval accuracy. Another consequence is that the memory $\mathbf{H}_t$ cannot be materialized and cached. Instead, one needs to maintain *all* the previous keys and queries in memory in order to compute the exponential of the dot products of the keys and queries, also referred to as the "KV Cache". A well known issue inherent to the self-attention mechanism, is that the KV Cache size (and per token computation) increases linearly with the sequence length.

**Gated linear attention.** Another way to limit interference between keys in the sequence is to learn when to "forget" information and remove it from the memory. This is the idea behind Gated Linear Attention (Yang et al., 2024), which improves stability and long-context performance through selective retention/forgetting of the information. Let $\boldsymbol{\alpha}_t, \boldsymbol{\beta}_t, \boldsymbol{\lambda}_t \in \mathbb{R}^{d_{qk}}$ be write-, read-, and decay-gates or, equivalently, broadcastable vectors. Gating is often implemented by learned affine maps and element-wise sigmoids. The update and reading rule are as follows:

$$\mathbf{H}_t = \operatorname{diag}(\boldsymbol{\lambda}_t)\,\mathbf{H}_{t-1} + \operatorname{diag}(\boldsymbol{\alpha}_t)\,\phi(\mathbf{k}_t)\,\mathbf{v}_t^\top, \tag{4}$$

$$\mathbf{y}_t = \big(\operatorname{diag}(\boldsymbol{\beta}_t)\,\phi(\mathbf{q}_t)\big)^\top \mathbf{H}_t. \tag{5}$$

Gated Linear Attention as well as other modern RNNs remove the normalizing constant and, instead, rely on normalizing layers such as LayerNorm (Ba et al., 2016) and RMSNorm (Zhang & Sennrich, 2019) in the network to stabilize training (Beck et al., 2025a).

**Sliding window attention (SWA).** Softmax attention maintains all past $(k_i, v_i)$ pairs, producing linear growth in memory and compute with $t$ due to the KV cache. Variants with a sliding window of size $w$ restrict the attention process to only the previous $w$ tokens, changing the memory and time complexity per-token from $O(S)$ to $O(w)$ complexity (Beltagy et al., 2020). This theoretically allows SWA architectures to handle arbitrarily large input sequences. In practice the receptive field of the model is limited to $O(lw)$ where $l$ is the number of SWA layers in the model. Moreover, it is unlikely that the theoretical receptive field is fully utilized in practice (Xiao, 2025).

**Hybrids between local attention and global softmax attention.** Through multi-turn interactions (Gehring et al., 2025), tool-use or long Chain-Of-Thought reasoning (Wei et al., 2023; DeepSeek-AI, 2025), the length which models have to process has grown from a few thousands of tokens to tens or hundreds of thousands of tokens. This motivates several recent works (OpenAI, 2025; Dong et al., 2024; NVIDIA, 2025; Ren et al., 2025) to consider new architectures whose computational cost grow less rapidly relative to sequence length than that of global softmax attention, while still performing well on long-context tasks. One such type of architectures are hybrids, for which most layers have a fixed state size like sliding-window or Linear Attention Layers, and the rest of the layers are global softmax attention layers. However, because those architectures still keep some global attention layers to remain competitive on long-context tasks, they also keep the $O(S)$ scaling in state size and FLOPs per token.

**Hybrids between linear attention and sliding window softmax attention.** Another kind of hybridization involves only component with a fixed state size, as considered by De et al. (2024) and Ren et al. (2025), who hybridize linear attention variants with sliding window attention. SWA paired with linear attention provides a natural split: the linear path maintains a compressed working memory with an unlimited receptive field; the windowed softmax path offers high-fidelity local reasoning.

| Unbounded memory | | Bounded memory | |
| --- | --- | --- | --- |
| transformer | local/global | xLSTM | SWAX |
| FFN | FFN | FFN | FFN |
| SA | SA | mLSTM | mLSTM |
| FFN | FFN | FFN | FFN |
| SA | SWA | mLSTM | SWA |
| FFN | FFN | FFN | FFN |
| SA | SA | mLSTM | mLSTM |
| FFN | FFN | FFN | FFN |
| SA | SWA | mLSTM | SWA |

Figure 3: We compare 4 different types of architectures, including 3 hybrid architectures:

(1) The transformer with vanilla self-attention (SA). Its complexity is prohibitive for long contexts lengths.

(2) This is circumvented by replacing some SA layers by sliding window attention (SWA) layers (Gemma Team, 2025; OpenAI, 2025).

(3) xLSTM (Beck et al., 2024) offers a memory with unbounded time horizon, albeit not as precise as SA for handling the recent context.

(4) SWAX is an hybrid architecture that includes both SWA layers and long-term memories layers, implemented with mLSTM memory cells.

Moreover, they demonstrate that, despite having *fewer* LA layers which are the only ones with an unlimited receptive field, the long-context performance of such hybrids is actually *higher* than that of a purely Linear Attention architecture. In particular, (De et al., 2024) investigated the impact of the size of the sliding window on validation perplexity. They found that longer windows yield better performance, making the choice of window size a purely a trade-off between performance and compute. However, they did not investigate the impact of the sliding window length on the *long-context* performance of the models.

## 3 HYBRID ARCHITECTURE DESIGN

In this work, we focus on hybrid architectures that alternate sliding window attention and linear RNNs. As a candidate for the linear RNN component, we choose the xLSTM (Beck et al., 2024), as this architecture has been scaled to models having up to 7B parameters and has shown strong performance in a wide variety of tasks. Importantly, fast and efficient Triton kernels are available (Beck et al., 2025b;a). xLSTM introduced two novel memory cells: the sLSTM with a scalar memory and the mLSTM with matrix memory. However, on language tasks the mLSTM cell shows superior performance over the sLSTM, which has been abandoned in the latest 7B xLSTM model. We follow this choice and rely solely on the mLSTM cell in our hybrid architecture. Subsequently, we use xLSTM and mLSTM interchangeably to refer to the same architecture.

There exist many ways to hybridize these two components (Wang et al., 2025). In our case, we adopt the simple design of inter-layer hybridization, which alternates between SWA layers and layers of linear attention. For the sake of simplicity, we adopt a 1:1 ratio meaning, that for every xLSTM layer there is one Sliding Window Attention layer. Figure 3 illustrates how the layers are interleaved in pure architectures and our SWA-xLSTM hybrid architecture. Most hybrids use window sizes between 128 as in OpenAI (2025) and 2048 as in De et al. (2024). We evaluate at intermediate lengths with sliding attention windows of lengths 128, 256, 512, 1024 and 2048. Finally, we evaluate a stochastic training procedure that aims at improving length-extrapolation. This training procedure stochastically chooses for each new batch either a short or a long window. In our experiments, we sample a window size of either 128 or 2048 with probability 0.5 for each length. A similar strategy was proposed by Zhang & Bottou (2025) in the context of the Memory Mosaic architecture. However, in their case, the stochastic attention mask was applied to a long-term memory layer. In constrast, we apply it to a Sliding Window Attention layer with the explicit goal of reducing over-reliance on the SWA layers for long-context recall.

## 4 EXPERIMENTS

### 4.1 EXPERIMENTAL SETUP

Our experiments focus on language modeling, with an emphasis on understanding the compromise between short-context and long-context recall performance. In particular, we investigate the impact of the SWA window size on long-context retrieval. For this purpose, we mainly rely on the needle-

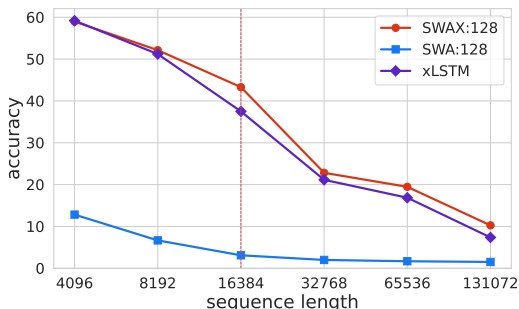

Figure 4: RULER needle-in-a-haystack average performance on varying sequence lengths for 1.4B models. For SWA and SWAX we indicate the sliding window size after the colon.

in-a-haystack tasks of the RULER benchmark (Hsieh et al., 2024). A common practice for models using global attention is to pre-train them on shorter sequence lengths like 4k or 8k to reduce the cost of the attention operation, and then fine-tuned in a second training stage on longer sequences to improve their long-context ability (Peng et al., 2023). In our case, we mainly focus on fixed-memory, fixed-compute architectures. Therefore, longer training sequences do not increase the required compute to attain a total training tokens target. We choose to train our models on a 16k sequence length from the start. Since we are interested in the capabilities of the model after standard pre-training, we do not perform any task-specific fine-tuning on long-context tasks. Except stated otherwise, our experiments use a model with 1.4 billion parameters. From our observations, it is at this size that models become able to perform recall on sequence lengths in the tens of thousands of tokens. However, to validate our method of stochastic window size at larger scale, we also evaluate models at the 7B parameter scale.

The 1.4B models have 24 blocks and a model dimension of 2048 while the 7B models have 32 blocks and a model dimension of 4096. In each block, the FFN is a gated MLP Liu et al. (2021) with Silu activation (Elfwing et al., 2017). For the SWA layers of the hybrids, we use Rotary Positional Embedding (RoPE) (Su et al., 2023) with a frequency $\theta$ of 10000, and 16 attention heads. All models are trained on 150 billion tokens following a warmup-cosine learning rate schedule with a peak learning rate of $3 \cdot 10^{-4}$ and a minimum learning rate of $3 \cdot 10^{-6}$. The batch size is $10^6$ tokens. Our training data mix consists mostly of web-data and code. Since we are most interested in the performance of models on long sequences, and our code data has, on average, 10 times longer documents than our web data, we report the validation perplexity on the code data subset of our data mix. For the MBPP (Austin et al., 2021) and HumanEvalPLus (Liu et al., 2023) pass@10 results, we use a sampling temperature of 0.8.

## 4.2 Hybrid *vs.* pure architectures

We start our investigation by reproducing the finding from (De et al., 2024) whose hybridization of Local Attention and Linear Attention variants improve performance across the board on both short-term reasoning and long-term recall tasks.

**Long-context performance.** Figure 4 shows that the pure SWA architecture performs poorly on long-context recall. This is expected because of its limited receptive field of 128 * 24 = 3072 tokens. More importantly, it confirms the counterintuitive finding from (De et al., 2024) that hybrids, despite having fewer global receptive field layers, outperform the pure variants in long-context recall. Intuitively, this is explained by the fact that, although the SWA layers have a limited receptive field, the softmax feature mapping allows them to better model local dependencies than an equivalent number of linear attention layers. Since most of the information necessary to predict the next token comes from local dependencies (Ruiz & Gu, 2025), a fully linear attention model dedicates most of its layers to modeling local dependencies and few layers to model long-term dependencies.

On the other hand, in SWA-LA hybrids, local dependencies are rather routed to the softmax local attention layers, which are more precise because of their direct access to recent history. As a consequence the linear attention layers specialize in modeling long-term dependencies, which the SWA

| Model | Transformer | xLSTM | SWA | SWAX | | | | |
|---|---|---|---|---|---|---|---|---|
| window length | n/a | n/a | 128 | 128 | 256 | 512 | 1024 | 2048 |
| FLOPs/token ($\times 10^9$) | 6.174 | 2.978 | 3.029 | 3.004 | 3.016 | 3.041 | 3.092 | 3.192 |
| val_PPL ↓ | **2.431** | 2.602 | 3.036 | 2.551 | 2.540 | 2.546 | 2.538 | **2.523** |
| HEplus/pass@10 ↑ | 14.63 | 12.80 | 13.41 | 12.20 | 12.80 | 14.02 | 14.63 | **15.24** |
| ARC-c ↑ | 30.90 | 28.93 | 31.25 | 29.27 | 30.13 | **31.76** | 30.30 | 29.79 |
| ARC-e ↑ | 66.43 | 65.79 | 65.58 | 65.75 | **67.82** | **67.82** | 66.89 | 67.15 |
| Hellaswag ↑ | 46.18 | 44.68 | 44.45 | 45.37 | 45.37 | 45.47 | 45.47 | **45.91** |
| MBPP/pass@10 ↑ | **31.40** | 22.60 | 23.80 | 24.20 | 26.80 | 28.40 | 29.20 | 28.80 |
| NaturalQuestions ↑ | 13.45 | 12.49 | 12.37 | **13.51** | 13.45 | 13.40 | 12.31 | 13.19 |
| PIQA ↑ | 73.99 | 73.39 | 73.99 | 73.83 | 74.05 | **74.48** | 73.67 | 73.99 |
| RACE.high ↑ | **37.19** | 33.65 | 33.25 | 33.56 | 35.71 | 35.71 | 35.11 | 35.99 |
| RACE.mid ↑ | **50.63** | 46.59 | 45.54 | 46.52 | 49.09 | 48.89 | 48.40 | 49.30 |
| SIQA ↑ | 42.48 | 40.23 | 41.61 | **42.53** | 42.02 | 41.71 | 41.91 | 41.71 |
| TriviaQA ↑ | **30.11** | 27.96 | 28.15 | 28.95 | 29.59 | 28.26 | 28.98 | 29.95 |
| Winogrande ↑ | 61.41 | 58.01 | **62.12** | 62.04 | 59.91 | 58.33 | 59.27 | 59.51 |
| **average** ↑ | **41.57** | 38.93 | 39.63 | 39.81 | 40.56 | 40.69 | 40.51 | **40.88** |

Table 1: Validation perplexity and accuracy on short-context reasoning and commonsense tasks. All models have 1.4B parameters. To compute the transformer FLOPs we use the training sequence length of 16384.

layers cannot model due to the limited window size. This highlights the impact that the window size can have on how much supervision the linear attention layers receive.

**Short-context performance** Table 1 shows that the performance of hybrid models on short-context reasoning benchmarks is higher than that of a xLSTM and also slightly higher than that of a pure SWA architecture. This further highlights the fact that for short contexts, hybrid models leverage the high precision of the softmax sliding window attention layers. Hybrid models therefore take the strong short-context performance of softmax attention, and the improved long-context recall ability of the Linear Attention layers.

### 4.3 IN SEARCH OF AN OPTIMAL WINDOW SIZE FOR HYBRIDS

Hereafter, we establish that windows that are too long actually hinder the Linear Attention layers from learning to model long-term dependencies during training. We hypothesize that this degradation is due to under-training of the linear attention layers on the long-context recall task. To validate this hypothesis, we train SWAX with varying window sizes in $\{128, 256, 512, 1024, 2048\}$ and test the models on both short context reasoning tasks and, more importantly, also on long-context recall tasks like RULER NIAH.

**Short-context performance.** De et al. (2024) experimented with different window sizes to find the optimal sliding window size. However, they only evaluated the different window sizes using the validation perplexity. Table 1 shows that, indeed, the hybrid with the largest softmax attention window (SWAX:2048) has the best performance from the validation perplexity point of view.

However, raw perplexity is not sufficient to accurately predict performance in downstream tasks, and especially not in long-context modeling tasks (Fang et al., 2024). Thus, we also evaluate the impact of the window size on short-context reasoning and common sense benchmarks and on long-context retrieval tasks from the RULER benchmark. Table 1 shows that on short-context reasoning benchmarks all window sizes except the shortest one, 128, give similar results, with the best performing hybrid being the one trained with the longest window size of 2048. The worse performance of the shortest window of size 128 is understandable as most prompts, even from those relatively short reasoning benchmarks, do not fit within a sliding window of 128 tokens.

**Long-context performance.** Figure 5 shows that once tested on longer sequences, the performance of the hybrid trained with a window size of 2048 drops the most. On the other hand, the SWAX models trained with shorter window sizes like 128, 256, and 512, maintain better performance even up to sequence lengths of 65k and 131k tokens. On the NIAH single task, SWAX models with a shorter window have around 30% recall accuracy at 131k sequence length, while the

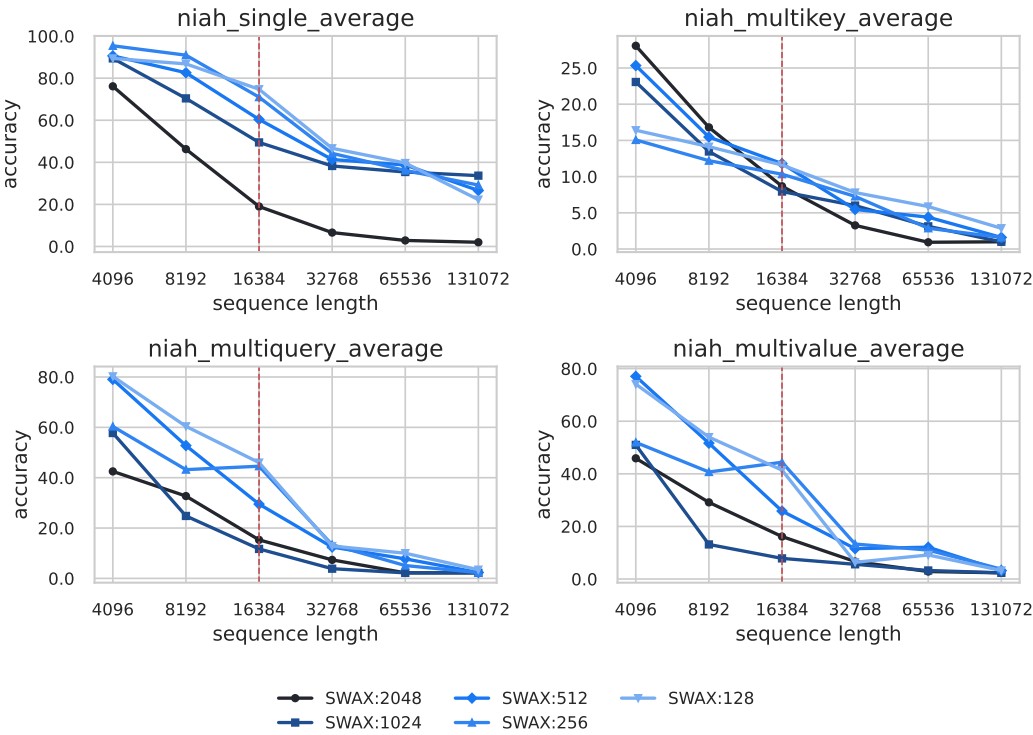

Figure 5: RULER NIAH subtasks accuracy for 1.4B SWAX models with different window sizes

SWAX with a window of 2048 has near 0% recall. Even the shortest sliding window of size 128, which consistently underperformed the longest ones in terms of PPL and short-context reasoning, significantly outperforms the model with the 2048 window length on all RULER NIAH tasks.

As shown in Figure 6, averaging over all sequence lengths and NIAH tasks, the SWAX model with a window size of 128 actually performs the *best* out of all the window sizes we tested. In particular, it outperforms the 2048 window size by 16 accuracy points. In other words, the SWAX with the shortest window has a recall *88.9% higher* than the SWAX with the longest window. The most likely cause for this phenomenon is that during training, most of the dependencies to model fall inside the 2048 tokens window.

Therefore, during pretraining, it was advantageous for the model with a window of 2048 to use the more precise softmax attention from the sliding window rather than having to rely on the less precise Linear Attention layers to model most dependencies. However, once tested on longer sequence length where the dependencies are outside of the window length, the model does not extrapolate since it never learned to rely on the Linear Attention layers to do long-context modeling.

On the other hand, the models with shorter windows *had to* rely on the Linear Attention layers to propagate information since many dependencies fell outside of the sliding window. We give further evidence in Appendix A which further indicates that this is indeed the reason for the poor long-context performance of hybrids with long sliding windows.

All these results show that, contrary to previous belief, longer sliding windows do not always provide better performance and can even have a *negative* impact when extrapolating to tasks beyond the sliding window size and training sequence length. On the contrary, shorter window sizes push the Linear Attention layers, that have a global receptive field, to receive more supervisory signal and specialize in long-context dependencies. Overall, shorter sliding windows allow the model to better extrapolate to tasks beyond the sliding window size and even far beyond the training sequence length. This also means that shorter windows are not just a way of reducing computational cost or maximize hardware utilization as was often thought to be the case, as in Arora et al. (2025) and De et al. (2024).

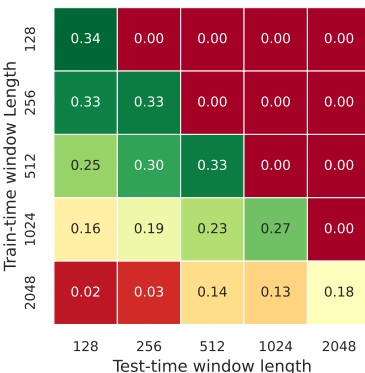

Figure 6: average NIAH accuracy of 1.4B SWAX models depending on their train and test time window sizes.

Table 2: Validation Perplexity and accuracy on downstream tasks. Stochastic models use $w = 2048$ by default and switch to $w = 128$ with probability $p = 0.5$ at 1.4B scale and $p = 0.75$ at 7B scale.

| model – parameters | SWAX 1.4B | | | SWAX 7B | | |
|---|---|---|---|---|---|---|
| train window length | 128 | stochastic | 2048 | 128 | stochastic | 2048 |
| test window length | 128 | 2048 | 2048 | 128 | 2048 | 2048 |
| val_PPL ↓ | 2.551 | **2.502** | 2.523 | 2.291 | **2.272** | 2.283 |
| HEplus/pass@10 ↑ | 12.20 | 12.80 | 15.24 | 24.39 | 24.39 | 26.83 |
| ARC-c ↑ | 29.27 | 30.82 | 29.79 | 40.77 | 40.86 | 41.55 |
| ARC-e ↑ | 65.75 | 68.71 | 67.15 | 75.05 | 74.46 | 74.80 |
| Hellaswag ↑ | 45.37 | 45.51 | 45.91 | 53.34 | 53.59 | 53.69 |
| MBPP/pass@10 ↑ | 24.20 | 30.60 | 28.80 | 44.60 | 47.20 | 45.40 |
| NaturalQuestions ↑ | 13.51 | 12.47 | 13.19 | 22.21 | 23.45 | 23.04 |
| PIQA ↑ | 73.83 | 74.43 | 73.99 | 77.26 | 77.97 | 76.55 |
| RACE.high ↑ | 33.56 | 35.91 | 35.99 | 37.65 | 38.99 | 39.88 |
| RACE.mid ↑ | 46.52 | 48.33 | 49.30 | 52.65 | 54.32 | 54.80 |
| SIQA ↑ | 42.53 | 41.86 | 41.71 | 43.55 | 44.22 | 42.78 |
| TriviaQA ↑ | 28.95 | 28.99 | 29.95 | 46.20 | 47.15 | 46.80 |
| Winogrande ↑ | 62.04 | 59.27 | 59.51 | 67.88 | 67.64 | 65.75 |
| average ↑ | 39.81 | 40.81 | **40.88** | 48.80 | **49.52** | 49.32 |

## 4.4 DIFFERENT WINDOW SIZES AT TRAIN AND TEST TIME

We now explore a training strategy allowing for a large window size at test time, to have the best reasoning performance possible, while still being trained such that the Linear Attention layers for long-term dependencies and extrapolate to longer sequences.

**Length extrapolation.** As a preliminary analysis, we first evaluate the performance of models when tested with a different window size than the one used at training time. Figure 6 shows that, as expected, naively extending the window size beyond its training length results in catastrophic collapse. This is a common phenomenon in softmax attention with RoPE which is used in the SWA layers (Peng et al., 2023). On the other hand, windows of size 1024 and less show little degradation when reducing their train-time window size by half. Overall, models need to be trained on large windows sizes to be able to use large windows during testing. At the same time, we cannot allow the models to over-rely on the long softmax attention windows since those do not perform well on very long-context tasks.

**Stochastic window size.** To solve this dilemma, we introduce a training procedure that, throughout the training, stochastically alternates between a large window size and a small window size. Our hypothesis is that this will prevent the model from over-relying on the SWA layers, while still making the model capable of using the larger window size at test time.

Moreover, to validate our experiments at a larger scale, we also train 7B parameter models using the same experimental setup as for the 1.4B models. For the 1.4B experiments, at each new batch of data, we set the window size to 128 with probability $p = 0.5$ or leave the default window size of 2048. At 7B scale, we use a slightly higher probability $p = 0.75$ of sampling the short window. We provide an ablation for the value of $p$ in Appendix B. Finally, to force the model to make better use of the larger test-time window of size 2048, we anneal the stochastic training procedure by not sampling the smaller window size anymore for the last 10% of training. We find that this short period of fixed windows at the end of the training significantly helps short-context performance without degrading long-context performance. We provide an ablation of the annealing in Appendix B.

Table 2 shows how training with a stochastic window size alternating between 128 and 2048 and annealing gives a short-context performance comparable to or even better than training with a fixed window size of 2048. In particular, at both 1.4B and 7B scales, stochastic training gives considerably better short-context performance than a fixed-sized window of 128. From a validation perplexity perspective, the stochastic window size outperforms all models trained with fixed window sizes at

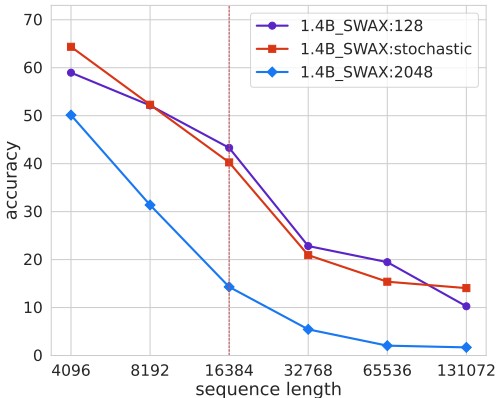

Figure 7: Average RULER NIAH accuracy of 1.4B SWAX models with different window sizes.

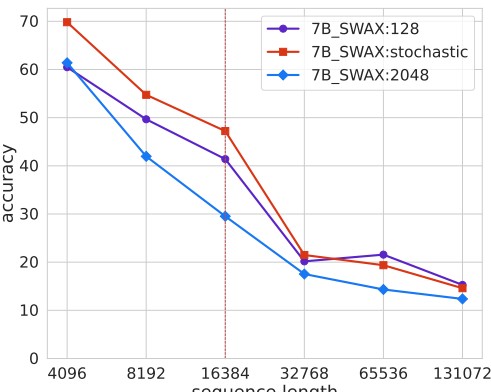

Figure 8: Average RULER NIAH accuracy of 7B SWAX models with different window sizes.

all parameter scales. Therefore, training with a stochastic window size and testing with a longer window yields better results on short-context tasks compared to a short window at both train and test time.

Compared to training with a fixed long sliding window of size 2048, stochastic training gives comparable performance at the 1.4B scale and even slightly superior performance at the 7B scale on short-context reasoning tasks. This indicates that indeed, even if during training the model has seen the longer window size only part of the time, it is still able to take advantage of the longer window size for short-medium context reasoning tasks.

**Long-context performance of the stochastic training.** We evaluate the stochastically trained SWAX models with annealing on the last 10% of the training. This is to ascertain this strategy gives a performance on long-context tasks as good as short-window variants.

Figures 7 and 8 show that on RULER's long-context recall tasks, the models trained with a stochastic window size and annealing perform on par or better than the model trained with the short window of 128, and drastically better than the hybrid model trained with a window of 2048 tokens. For instance, at 1.4B parameter scale depicted in Figure 7, the stochastic training SWAX model performs similarly to the short-window SWAX. Figure 8 shows that stochastic training also improves the long-context performance of models at 7B scale. At 7B scale, compared to using a fixed sliding window of size 2048, the stochastic training gives much better retrieval accuracy at all sequence lengths. Furthermore, just as at 1.4B scale, stochastic training gives similar or even superior long-context performance compared to using a short window throughout training. Overall, a stochastic window size at training time maintains all the benefits of having a short window for long-context recall, and most if not all of the benefits of a longer window for short/medium-context reasoning tasks.

Moreover, this result further confirms that the poor performance of hybrids trained on long windows is not intrinsically due to a long sliding window at test time. Instead, these results show that the poor length-generalization of hybrids with long sliding windows is due to the training procedure. Indeed, if the model is allowed to use the long sliding window throughout training, it will over-rely on the more precise softmax attention of the sliding window even for recall tasks, which will not extrapolate to longer sequence lengths. It will under-utilize the Linear Attention Layers for the long-term recall task. On the contrary, if during training the model is not allowed to rely on the long window, and is instead stochastically forced to use a shorter window, then the linear attention will be required on the medium/long-term recall tasks. Since, essentially, this amounts to stochastically reducing the capacity of the model to make it more robust, this can be seen as a form of dropout (Srivastava et al., 2014) on the attention mechanism.

**More long-context benchmarks.** The Ruler tasks are artificial long context benchmarks, built to be sensible to model variants. We also evaluated SWAX on more realistic tasks from several families of benchmarks:

- **LongBench** (Bai et al., 2024a) is a bilingual, multitask, and comprehensive benchmark to evaluate the long context understanding capabilities of LLMs. It is composed of six categories and twenty one different tasks. It evaluates long-text application scenarios such as single-document QA, multi-document QA, summarization, few-shot learning, synthetic tasks and code completion.
- **Babilong** (Kuratov et al., 2024) This benchmark extends the Babi tasks (Weston et al., 2015), it is designed to evaluate long-term memory and reasoning capabilities of LLMs. Babilong uses long input sequences, making it suitable for evaluating models with advanced memory architectures. The main categories are: single book reasoning, memory and retrieval and temporal and spatial reasoning.
- **LongBench2** (Bai et al., 2024b) builds on the original LongBench, expanding both the scale and diversity of tasks to better stress-test LLMs for real-world long-context scenarios.

The results are summarized in Table 3 (bottom). They show that the tasks are difficult on average for such small models. Stochastic training outperforms fixed-size training in some settings, but for others, the long context (2048) performs better. In Table 5, Appendix D we present the average performance of the LongBench summarization and question-answering tasks, where the same conclusion holds.

| Task family | Training SWA size | | |
| --- | --- | --- | --- |
| | 128 | stochastic | 2048 |
| Longbench | 10.67 | **11.79** | 11.14 |
| Longbench2 | 22.99 | **27.26** | 22.65 |
| Babilong | 4.68 | **9.35** | 7.29 |
| Gated DeltaNet | | | |
| Ruler multiquery NIAH | 40.58 | **40.90** | 28.80 |
| Longbench | 9.14 | 9.94 | **11.42** |
| Babilong | 3.74 | 7.21 | **8.95** |

Table 3: Other long-context benchmarks on our SWAX architecture with 1.4B parameters. The results are averaged over all tasks within a family. Top: xLSTM linear attention memory, bottom: Gated DeltaNet linear attention. For Gated DeltaNet we also report the averaged performance of Ruler multiquery NIAH, averaged over sequence sizes 4096, 8192 and 16384.

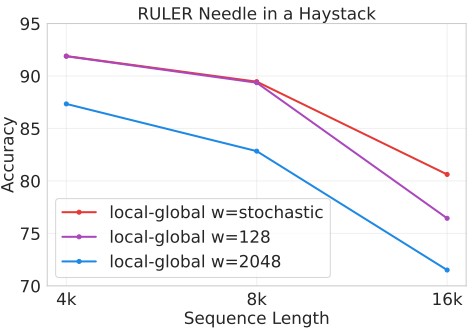

Figure 9: RULER Needle in a Haystack average accuracy for 1.4B local:global models over NIAH single1, 2, 3, Multiquery and Multikey, from 4k sequence length up to 16k.

### 4.5 EXPERIMENTS WITH GATED DELTANET AND FULL ATTENTION

The Gated DeltaNet (Yang et al., 2025) is a state-of-the-art hybrid architecture that combines gating and delta-update rule to update the linear attention memory. To assess whether our observations apply to other linear attention architectures, we replace the xLSTM component in the previous experiments with a Gated DeltaNet layer (we call this combination SWAX/GDN). We did not change any training setting, and ran the same long-context benchmarks. Table 3 (top) shows that hybrid architectures with Gated DeltaNets also benefit from the stochastic training.

To further validate our findings, we also trained and evaluated local-global architectures which alternate SWA layers with full attention layers. This setting, which keeps some amount of full attention layer in the architecture, is commonly used by SoTA open source models like OpenAI (2025) or Singh et al. (2026). As shown in figure 9, using larger sliding windows during training also degrades the long-context performance of local-global architectures with full attention. This reinforces the fact that our findings generalize to many kind of hybrid attention architectures.

## 5 CONCLUSION

Through an empirical analysis of hybrid architectures, we evidence the counter-intuitive fact that shorter sliding windows lead to better length-extrapolation on retrieval tasks. Moreover, we intro-

duce a training procedure that stochastically changes the window size throughout training. This training procedure offers a strong performance on short-context tasks (enabled by longer sliding windows) and the length-extrapolation ability of Linear Attention layers, enabled by shorter windows at training time.

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

# SUPPLEMENTARY MATERIAL

## A  RESULTS OF PURE SWA MODELS

In section 4.3 we hypothesize that the worse performance of SWAX models with long windows comes from the model utilizing the SWA layers instead of the xLSTM layers. To further confirm this hypothesis, we train a 1.4B pure SWA model with a window size of 2048 and compare its performance to the SWAX model with the same window size. If the hypothesis that the SWAX model relies on the SWA layers for recall is valid, then we expect its performance to be similar to that of a pure SWA architecture.

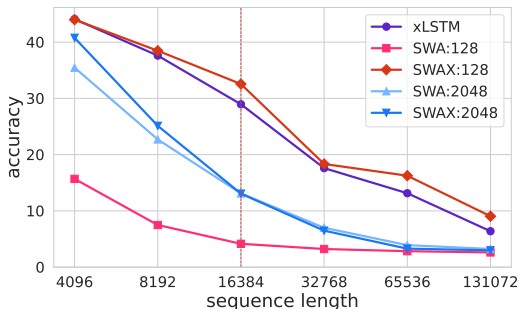

Figure 10: Average accuracy of 1.4B parameter models over all RULER NIAH tasks.

Figure 10 shows that indeed, the SWAX model trained with a window of 2048 performs very similarly to the pure SWA architecture. On the other hand, the accuracy the SWAX model trained with a window of 128 is dissimilar from that of the pure SWA model with a window of 128. This further evidences that low long-context performance of hybrid models with long windows comes from the model over-relying on the SWA layers for long-context recall instead of using the xLSTM layers.

## B  STOCHASTIC SAMPLING PROBABILITY AND ANNEALING OF STOCHASTICITY

In this experiment we perform an ablation on the stochastic sampling probability $p$ its schedule during training for the two model sizes 1.4B and 7B we consider in our study. Table 4 shows that, at 7B scale, a higher probability of sampling the small window during training is necessary to significantly improve short and long context performance compared to the probability of 0.5 which worked at 1.4B scale. Looking at the impact of annealing, i.e. using a stochastic window size for the first X% of training, at both 1.4B and 7B scale, annealing improves short-context performance compared to keeping the stochasticity until the end of training. At 7B scale, the annealed SWAX model even performs better on short-context than the SWAX model trained with a fixed window size of 2048. In terms of long-context performance, compared to keeping the stochasticity until the end of training, annealing slightly degrades the long-context performance at 1.4B scale but keeps or even slightly improves long-context performance at 7B. We believe that exploring different annealing procedures might provide even better short-context performance improvements while — at the same time — keeping good long-context performance.

## C  BENCHMARKS

**Code generation**  We use two benchmarks that evaluate the code generation capabilities of AI models: HumanEval+ and MBPP.

- The HumanEval+ (Liu et al., 2023) benchmark is an extension of HumanEval (Chen et al., 2021), which is designed to evaluate the functional correctness of code generated by AI models.

| model parameters | xLSTM 7B | SWAX 7B | | | | | | SWAX 1.4B | |
|---|---|---|---|---|---|---|---|---|---|
| train-time window
test-time window | NA
NA | 128
128 | $p$=0.9
2048 | $p$=0.75
2048 | $p^{90\%}$=0.75
2048 | $p$=0.5
2048 | 2048
2048 | $p$=0.5
2048 | $p^{90\%}$=0.5
2048 |
| **niah_single** | 61.20 | 62.43 | 58.99 | **63.36** | 63.20 | 55.27 | 53.46 | 62.61 | 61.59 |
| **niah_multiquery** | **44.18** | 34.78 | 35.95 | 32.72 | 32.23 | 17.94 | 21.62 | 30.85 | 27.42 |
| **niah_multikey** | 10.14 | 9.96 | 13.32 | 17.23 | **17.86** | 14.34 | 12.29 | 14.52 | 12.19 |
| **niah_multivalue** | **39.28** | 26.11 | 23.55 | 27.32 | 27.56 | 19.48 | 17.30 | 30.63 | 27.63 |
| **niah_average** | 37.19 | 34.76 | 34.55 | 37.73 | **37.87** | 30.78 | 29.52 | **36.61** | 34.55 |
| **HEplus/pass@10** | 25.00 | 24.39 | 25.61 | 23.17 | 24.39 | 21.95 | **26.83** | 13.41 | 12.80 |
| **arc-c** | 37.42 | 40.77 | 40.00 | 40.86 | 40.86 | 40.09 | **41.55** | 32.45 | 30.82 |
| **arc-e** | 73.32 | **75.05** | 74.76 | 74.50 | 74.46 | 74.63 | 74.80 | 67.23 | 68.71 |
| **hella** | 52.95 | 53.34 | 53.34 | 53.48 | 53.59 | 53.47 | **53.69** | 45.61 | 45.51 |
| **mbpp/pass@10** | 43.80 | 44.60 | 43.60 | 42.80 | **47.20** | 45.80 | 45.40 | 28.00 | 30.60 |
| **nq** | 22.12 | 22.21 | 23.12 | 23.16 | **23.45** | 22.58 | 23.04 | 12.95 | 12.47 |
| **piqa** | 76.93 | 77.26 | 76.71 | **78.07** | 77.97 | 77.31 | 76.55 | 74.32 | 74.43 |
| **race.high** | 37.02 | 37.65 | 37.71 | 38.74 | 38.99 | 38.74 | **39.88** | 34.82 | 35.91 |
| **race.mid** | 52.85 | 52.65 | 54.11 | 54.32 | 54.32 | 53.90 | **54.80** | 48.26 | 48.33 |
| **siqa** | 43.96 | 43.55 | 44.06 | 44.01 | 44.22 | **44.37** | 42.78 | 41.91 | 41.86 |
| **tqa** | 46.39 | 46.20 | **47.44** | 46.90 | 47.15 | 46.55 | 46.80 | 29.35 | 28.99 |
| **wino** | 66.06 | 67.88 | **68.51** | 67.32 | 67.64 | 66.77 | 65.75 | 59.20 | 59.27 |
| **short-average** | 48.15 | 48.80 | 49.08 | 48.95 | **49.52** | 48.85 | 49.32 | 40.62 | **40.81** |

Table 4: NIAH and downstream tasks accuracy for 7B models. p indicates the probability of using a window of 128 for a batch, otherwise using a window of 2048. $p^{90\%}$ indicates annealing, i.e., only doing the stochastic window size for the first 90% of the training and then using a fixed window size of 2048 for the rest of training. NIAH single and multikey results are the average overall all 3 sub-tasks for each.

- MBPP (Austin et al., 2021) is designed to evaluate the code generation abilities of AI models, particularly for Python programming tasks.

**Common sense and general reasoning.** We use benchmarks consisting of question-answer or multiple-choice questions designed to evaluate the common sense reasoning abilities of AI models, particularly in the context of natural language understanding: HellaSWAG (Zellers et al., 2019), ARC (Clark et al., 2018), PIQA (Bisk et al., 2019), SIQA (Sap et al., 2019), Winogrande (Sakaguchi et al., 2019), NaturalQuestions (Kwiatkowski et al., 2019), RACE (Lai et al., 2017) and TQA (Joshi et al., 2017).

## D  LONGBENCH SUMMARISATION AND QA TASKS PERFORMANCE

We can report the averages for all the summarization and question answering tasks, we observe that training with stochastic SWA size outperforms the fixed-size training windows.

| Task | SWAX:128 | SWAX:2048 | SWAX:stochastic |
|---|---|---|---|
| Longbench summarization tasks (average) | 10.570 | 11.152 | **12.429** |
| Longbench QA tasks (average) | 5.203 | 6.112 | **7.257** |

Table 5: Average performance for summarization and question-answering LongBench tasks

- The LongBench summarization tasks are: longbench_gov_report,longbench_qmsum, longbench_multi_news and longbench_vcsum.
- The LongBench QA tasks, which include single-QA and multi-QA tasks, are: longbench_hotpotqa, longbench_2wikimqa, longbench_musique, longbench_dureader, longbench_narrativeqa, longbench_qasper, longbench_multifieldqa_en and longbench_multifieldqa_zh.

