# OpenReview forum: "Short Window Attention Enables Long-Term Memorization"
_ICLR.cc/2026/Conference — ICLR 2026 Poster_

### Official Review · Reviewer_bBjL · 2025-10-29

**Soundness:** 3
**Presentation:** 4
**Contribution:** 3
**Rating:** 8
**Confidence:** 4

**Summary:**

This paper empirically evaluates how the window size of sliding window attention affects the long-context performance, when mixing sliding window attention with xLSTM linear RNN layers in a hybrid architecture. The authors find that short window attention encourages the model to better train xLSTM, leading to better long-context performance. However, short window attention may lead to worse short-context performance. The authors then propose a simple strategy to randomize the window size in training, achieving the best of both worlds.

**Strengths:**

1. The paper is well-written, clear, and easy to follow. The paper provides an interesting and practically insightful observation that short window attention leads to better long-context performance in hybrid models mixing window attentions and RNN layers.

2. The authors provide a simple adaption to train a model with good short-context and long-context performance.

**Weaknesses:**

1. The paper would be stronger if it evaluates more than one types of RNN layers to show that the findings are robust against different RNN architectures.

2. The findings are based on models without fine-tuning with long-context data. It is unclear whether the results continue to hold if the models are further fine-tuned.

**Questions:**

1. Could the authors comment on the generalizability of the observation to other RNN architectures?

2. Could the authors comment on whether the results continue to hold if the models are further fine-tuned?

---

> ### Author Response · Authors · 2025-11-26
>
> We thank the reviewer for their detailed feedback. We appreciate that our results are found insightful and our method clear and simple. We address some concerns below.
>
> 1. **Applicability to other models/RNNs:**
> We agree with the reviewer on this point. see global comment 2.
>
> 2. **Finetuning:**
> We agree with the reviewer that it is unclear whether or not the results continue to hold if the models are further finetuned on long-context data. Our aim with pre-training at 16k sequence length, which is higher than the more common 4k or 8k pre-training sequence lengths, was to show that these results already hold in a relatively long-context data regime. Would the reviewer find fine-tuning with sequence lengths up to 64k satisfactory to confirm whether these results hold under long-context fine-tuning?

---

### Official Review · Reviewer_SVZm · 2025-10-31

**Soundness:** 3
**Presentation:** 3
**Contribution:** 2
**Rating:** 4
**Confidence:** 4

**Summary:**

The paper proposes SWAX, a hybrid model that alternates sliding‑window softmax attention (SWA) with xLSTM (linear RNN) layers, arguing that short attention windows during training force the model to use the RNN path as true long‑term memory. Empirically, for 1.4B and 7B models trained on 150B tokens at 16k sequence length, the authors find: (i) shorter windows (e.g., 128) yield better long‑context retrieval on RULER Needle‑in‑a‑Haystack (NIAH) than longer windows (e.g., 2048) when extrapolating up to 131k tokens, and (ii) a stochastic window schedule preserves long‑context gains while recovering short‑context quality at test time with a 2048 window.

**Strengths:**

- Clear, counter‑intuitive empirical result: In hybrids, short windows train better long‑term memory than long windows. The heat‑map in Fig. 6 (p.8) plus the degradation curves in Fig. 5 (p.7) make the effect easy to verify.
-  The stochastic window + brief annealing yields near‑best short‑context results while keeping long‑context robustness. This is a practical recipe others can adopt with minimal code churn.

**Weaknesses:**

- Long‑context evaluation is narrow: RULER NIAH is synthetic. The paper lacks real‑task long‑context evaluations.
- On the short‑context suite, the Transformer baseline still leads on average. The paper should contextualize this gap with wall‑clock throughput and memory at inference.
- Design space under‑explored: The hybrid is fixed to a 1:1 SWA:xLSTM interleave and xLSTM only. It would be more valuable if more results on other setup could be added.

**Questions:**

N/A

---

> ### Author Response · Authors · 2025-11-26
>
> We thank the reviewer for their detailed feedback. We appreciate that our results are found to be counter-intuitive and our method simple, practical and effective.We address a few concerns below.
>
> 1. **Limited Evaluation (NIAH focus):**
> We agree with the reviewer on this point. See global comment 1.
>
> 2. **Transformer results:**
> We agree with the reviewer that this needs more clarification. The results of the quadratic time-complexity and non-fixed hidden-state transformer baseline were added to put the other results in context. We underline that our contribution does not claim to, nor aim at beating the transformer baseline. The transformer belongs to a different complexity class compared to the other architectures we evaluate. We will add wall-clock throughput and memory and inference indications in the revised version.
>
> 3. Applicability to other models/RNNs
> We agree with the reviewer on this point. See global comment 2.
>
> Would reviewer SVZm also find additional experiments with different hybridization ratios like 3:1 SWA:xLSTM valuable to further confirm the generality of our findings and method?

---

### Official Review · Reviewer_pAFQ · 2025-11-01

**Soundness:** 3
**Presentation:** 3
**Contribution:** 3
**Rating:** 6
**Confidence:** 3

**Summary:**

The paper studies hybrid models that interleave sliding-window softmax attention (SWA) with linear RNN layers (xLSTM/mLSTM) in a 1:1 pattern. The core, counter-intuitive claim is that short SWA windows improve long-context recall because they push the model to train its recurrent memory instead of relying on local softmax; conversely, long windows hurt length extrapolation. The authors also propose stochastic window training—randomly switching between short and long windows during pretraining with an anneal near the end—to regain short-context quality without giving up the long-context gains. Empirically, they evaluate mainly on RULER “needle-in-a-haystack” tasks and a suite of short-context benchmarks;

**Strengths:**

1) Clear, testable capacity-allocation hypothesis. Long windows allow SWA “absorb” the learning signal; short windows force the xLSTM to learn long-term memory. The paper probes this across window sizes and model scales.

2) Simple, practical training recipe. Stochastic window training with a short end-of-training anneal; task-agnostic and effective at 1.4B and 7B.

3) Comprehensive comparisons. Side-by-side results for pure (xLSTM, SWA) vs. hybrid models, compute-aware reporting (FLOPs/token; Table 1), and stringent length-extrapolation stress tests (RULER NIAH up to 131k).

**Weaknesses:**

1) The pretraining mix is described only at a high level (“mostly web and code”); perplexity is reported on a code subset, but sources, filters, licenses, and exact proportions are not disclosed. This reduces reproducibility and external validity.

2) RULER NIAH—primarily a needle-retrieval test—is the centerpiece, but broader long-context reasoning is underrepresented (multi-hop QA over long documents, extended chain-of-thought, codebase-level tasks). The generality of the findings remains unclear. A stronger case would include evaluations across varied tasks, e.g., Babilong and long open-QA benchmarks (HotpotQA and MuSiQue in full-wiki mode, LongBench).

3) The experiments are exclusively based on the xLSTM as the linear RNN component. A brief discussion or experiment on whether this phenomenon holds for other popular models, including neural memory (for example, Titans[1]), would significantly increase the generalizability and impact of the findings.

[1] Ali Behrouz and Peilin Zhong and Vahab S. Mirrokni. Titans: Learning to Memorize at Test Time, arXiv:2501.00663

**Questions:**

1) How many random seeds were run per configuration? Please report mean ± standard deviation across seeds for the key metrics in Table 1 and Figures 5–8 (and include the per-seed results in an appendix or supplementary file). If only a single seed was used for any result, please state this explicitly.

2) Could you specify the full optimization setup: the optimizer used (e.g., AdamW), exact β₁/β₂ (and ε) values, weight-decay coefficient, gradient-clipping norm and threshold, all dropout rates?

---

> ### Author Response · Authors · 2025-11-26
>
> We thank the reviewer for their detailed feedback. We appreciate that you found our method simple and practical, our evaluation suite comprehensive and our method effective at both 1.4B and 7B scale.
> We address your concerns below.
>
> 1. **Datamix:**
> The datamix is about ⅔ DCLM (https://arxiv.org/abs/2406.11794), ⅓ code from github, and a series of specialized corpuses (math, wikipedia definitions), that add less than 1% of the datamix. We are sorry we can’t disclose precise details.
>
> 2. **Limited Evaluation (NIAH focus):**
> We agree with the reviewer on this point. See global comment 1.
>
> 3. **Applicability to other models/RNNs:**
> We agree with the reviewer on this point. See global comment 2.
>
> 4. **Number of seeds Used:**
> Due to the runtime cost of training models from scratch, only 1 seed is used for each run. We agree with the reviewer that this information should clearly be stated explicitly and will do so in the revised version.
>
> 5. **Training hyperparameters:**
> The Optimizer used is AdamW with β₁/β₂ (and ε) respectively equal to 0.9, 0.95 and 10^-8, weight decay equal to 0.1 and gradient clipping using l2 norm and threshold set to 1, dropout probability set to 0 (not counting stochastic window training as dropout).

---

### Official Review · Reviewer_4vk4 · 2025-11-03

**Soundness:** 3
**Presentation:** 2
**Contribution:** 2
**Rating:** 2
**Confidence:** 3

**Summary:**

The paper proposes **SWAX**, a hybrid that interleaves sliding-window softmax attention (SWA) and xLSTM layers. Core claims: (i) **shorter** SWA windows surprisingly yield **better long-context recall** because they force the RNN path to learn long-term memory; (ii) a **stochastic window** training schedule (occasionally using a small window such as 128, otherwise 2048, with an anneal in the last 10% of training) recovers short-context performance while preserving long-context extrapolation. Evaluations use a 1.4B and 7B model trained on 150B tokens at 16k sequence length; long-context performance focuses on RULER NIAH, with downstream short-context benchmarks reported.

**Strengths:**

The stochastic window schedule (sampling 128 vs 2048; annealing off in the last 10% of training) is simple, inexpensive to try, and consistently improves length extrapolation without sacrificing short-context quality at both 1.4B and 7B scales. This is well demonstrated in Figs. 7–8 and Table 2.

**Weaknesses:**

1. **Limited and unbalanced evaluation of “long-context” ability.**
   The central claim rests mostly on **RULER Needle-in-a-Haystack (NIAH)** accuracy; other long-context behaviors (QA, summarization, codebase-level reasoning, tool-use traces, multi-hop retrieval) are not evaluated. As a result, it’s unclear whether the observed gains generalize beyond synthetic token recall. The paper itself emphasizes RULER NIAH and heatmaps across sequence lengths but lacks a broader long-context suite. This weakens external validity of the main claim.

2. **Confounds in the “short vs long window” story and insufficient controls.**
   The paper argues that **longer** windows under-train the RNN path and therefore **hurt** extrapolation; however, compute is not cleanly equalized across variants (Table 1 shows **FLOPs/token** rising with window size), and the training schedule/data mix may interact with window length (e.g., document length distribution skewed to code). Stronger controls (fixed total FLOPs, matched optimization schedules, varied data mixes, and tests at matched effective receptive fields) are needed to isolate causality. As is, the causal interpretation (under-training the RNN) is plausible but not conclusively established.

3. **Novelty is incremental relative to prior hybrid work; framing oversells.**
   Alternating xLSTM and local attention is now common; the **new ingredient** here is mainly the **stochastic window schedule** and the empirical observation that small windows train the long-term path better. The paper acknowledges closely related hybrids and prior window-size analyses (e.g., De et al.), and even positions the method as akin to **dropout** on the attention mechanism. Without broader/better-controlled evaluations or theoretical support, the contribution feels like a useful training tip rather than a substantial architectural advance.

**Questions:**

* Line 075: “**hybrids architecture**” → “**hybrid architectures**.”
* Line 198: “In **constrast**” → “In **contrast**.”
* Line 233:  “we also **train evaluate** models at the 7B parameter scale.” → “train **and** evaluate.”
* Table/figure captions could be clearer (e.g., units/averaging in Fig. 6 heatmap; spell out metrics).
* **Add broader long-context tasks** (e.g., LongBench-style QA/summarization/code comprehension) to test whether the NIAH gains translate to practical workloads.
* **Tighter causal tests**: (i) equalize total compute across window settings; (ii) vary document-length distribution; (iii) report how gradients/attention mass shift between SWA and xLSTM under different windows; (iv) include ablations on the anneal schedule (you mention some in Appendix B—bring the most informative ones into the main paper).

---

> ### Author Response · Authors · 2025-11-26
>
> We thank the reviewer for their detailed critique. We appreciate that the stochastic schedule is found to be "simple, inexpensive, and consistent." About the concerns:
>
> 1. **Limited Evaluation (NIAH focus):**
> We are busy adding more experiments, see global comment 1.
>
> 2. **Confounds and Causal Interpretation:**
> We will clarify that the optimization schedules are identical between all models presented there.
> The point regarding FLOPs/token increasing with window size is valid: there is a < 10% FLOP variation between different training window sizes. We chose instead to equalize the **token budget** for comparison because this has a higher impact on the results.
> Besides, it turns out that this confound actually strengthens our claim: typically, more FLOPs (longer windows) should lead to better performance, Our result shows that the computationally cheaper option (short windows) **also** yields better long-term memory.
> This suggests the gain is indeed due to the inductive bias from a shorter sliding window rather than compute budget. We will clarify this in the revised version.
>
> 3. **Novelty and Framing:**
> We acknowledged in the paper that alternating attention and RNNs is an established practice. We will reframe the contribution to emphasize that our primary finding is the training dynamics discovery (the interaction between window size and RNN utilization) and the practical stochastic schedule recipe, rather than a new architecture per se.
>
> We will also improve the editorial quality of the text.

---

### Author Response · Authors · 2025-11-26
**General reply**

We thank the reviewers for the time spent analysing our paper. Global comment for recurrent questions:

1. Limited Evaluation / focus on the needle-in-a-haystack (NIAH) benchmark:
We fully agree with the reviewers that NIAH is a necessary but insufficient proxy for fully evaluating long-context performance. We are planning to run the LongBench suite (including QA and Summarization) and will report these results as soon as possible.

2. Applicability to other models/RNNs:
Our analysis is limited to the xLSTM layer and confirming these findings with other architectures would show these findings generalize. We are trying to extend our experiments to the publicly available Gated DeltaNet architecture to confirm this. We will report on the progress, but it may be hard to achieve by Dec 3.

---

> ### Author Response · Authors · 2025-12-03
> **Additional results**
>
> We obtained results for additional benchmarks. They are summarized in the table:
>
> | Task average performance | SWAX:128 | SWAX:2048 | SWAX:stochastic |
> |--------|-------------|--------------|----------------|
> | longbench | 10.673 | 11.137  | **11.790** |
> | longbench2 | 22.986 | 22.653 | **27.264** |
> | babilong | 4.684 | 7.286 | **9.347** |
>
> In general, training with stochastic SWA size outperforms the fixed-size training windows by a significant margin (although absolute performance is relatively low, due to the small model scale).
> The table below focuses on a few tasks the reviewers were particularly interested in:
>
> | Task  | SWAX:128 | SWAX:2048 | SWAX:stochastic |
> |--------|-------------|--------------|----------------|
> | Longbench summarization | 10.570 | 11.152 | **12.429** |
> | Longbench QA (average) |  5.203 | 6.112 | **7.257**|
>
> So for summarization and question answering, the same conclusions hold.
>
> To show that our findings extend to other types of linear memory, we replaced the xLSTM layer with a Gated DeltaNet, another state-of-the-art linear attention variant. The results are below:
>
> | Task  | SWAX/GDN:128 | SWAX/GDN:2048 | SWAX/GDN:stochastic |
> |--------|-------------|--------------|----------------|
> | NIAH multiquery  | 40.583 |     | **40.900** |
> | longbench        | 9.138 |   | **9.939** |
> | babilong         | 3.743 |   | **7.206** |
>
> Unfortunately we were unable to obtain the results for a SWA context of 2048 and for longbench2 in time for the rebuttal. However, we observe that stochastic training is beneficial relative to a fixed window size of size 128.
>
> We updated the manuscript with these new results, and will add the missing ones when ready. We also carefully proofread the document to improve its editorial quality.

---

### Meta-Review · Area_Chair_8Put · 2026-01-11

**Summary:**

This paper contributes (1) a counter-intuitive but clearly demonstrated phenomenon in SWA (local attention) + linear-memory hybrids: shorter sliding windows improve length extrapolation on long-context retrieval, and (2) a simple stochastic window training procedure that largely restores short-context quality while retaining long-context robustness. Evidence is strong on RULER NIAH across lengths up to 131k, and the rebuttal adds additional long-context benchmarks (LongBench/2, Babilong) and an additional linear-memory backend (Gated DeltaNet).

However, concerns remain around breadth of “real” long-context reasoning evaluation, limited exploration of the hybrid design space, and the fact that results are single-seed and largely pretraining-only (no long-context fine-tuning study). Major concern is this focus heavily in pretrain regime, and it is unclear if effects persist after long-context fine-tuning or instruction tuning.

Despite some remaining gaps in breadth, this submission makes a compelling and practically important contribution: it isolates a counter-intuitive but important training dynamic in hybrid models (shorter sliding windows can improve long-context length extrapolation in hybrid SWA+linear-memory models). Rebuttal strengthens experiments with additional long-context benchmarks.

**Reviewer Concerns:**

Addressed

NIAH-only long-context evaluation: improved via LongBench/LongBench2/Babilong + QA/summarization averages showing the same trend.
xLSTM-only generality: partially addressed via Gated DeltaNet results
Missing training details: clarified optimizer/hparams, and explicitly stated single-seed.

Still outstanding
1. External validity beyond retrieval: added benchmarks help, but overall evaluation still leans toward retrieval/memory; codebase-level long-context tasks, tool-use trajectories, etc., remain untested.
2. Compute/control confounds: For the main empirical claim (“shorter SWA windows leads to better length-extrapolation in this hybrid, compute/FLOPs confound is not a big issue to me, cause the effect is opposite of “more compute helps”. For the causal mechanism claim (“long windows under-train the xLSTM path / attention ‘steals’ learning signal”), confounds are still a medium issue.
3. Design space: fixed 1:1 interleave; limited ablations on hybrid ratios / placements.
4. unclear if effects persist after long-context fine-tuning or instruction tuning.

**Reviewer Scores:**

pAFQ (rebuttal directly addresses reproducibility and generality requests) and 4vk4 (broader eval) may increase scores,

---

### Decision · Program_Chairs · 2026-01-26

Accept (Poster)